# Quantification of Groundwater Vertical Flow from Temperature Profiles: Application to Agua Amarga Coastal Aquifer (SE Spain) Submitted to Artificial Recharge

**José Antonio Jiménez-Valera *** , **Iván Alhama *** and **Emilio Trigueros**

Mining and Civil Engineering Department, Universidad Politécnica de Cartagena, Paseo Alfonso XIII, 52, 30203 Cartagena, Spain
* Correspondence: jose.jvalera@upct.es (J.A.J.-V.); ivan.alhama@upct.es (I.A.)

**Abstract:** The Agua Amarga coastal aquifer has experienced different anthropic interventions over the last 100 years. Since 2008, groundwater abstractions along the coastline to supply the Alicante I and II desalination plants have been combined with artificial recharge. This measure, consisting of seawater irrigation over the salt marsh, has reduced the impact on the piezometry and surface ecosystems. The study of the effect of this measure on groundwater flow is addressed by applying an inverse problem protocol to temperature profiles monitored in a piezometer located inside the recharge area. Information on monthly seawater irrigation volumes, rainfall, and average air and seawater temperatures have also been gathered as input data to quantify vertical flow. An upward flow component for the period 2010–2022 that varies between $2 \times 10^{-9}$ and $7.5 \times 10^{-7}$ m/s has been found. These values decrease near the surface, where the flow is mainly horizontal.

**Keywords:** groundwater flow; numerical simulations; artificial recharge; inverse problem; temperature–depth profile time series (TDPTS)

## 1. Introduction

The use of groundwater temperature as an indicator of processes related to flow has been broadly applied [1–6]. The physics on which it is based is the coupled processes of flow and heat transfer due to convection and advection phenomenon in porous media [7,8]. Among the mathematical techniques used for this purpose are numerical methods, the inverse problem, and analytical solutions. Suzuki [9] was the first author to propose an analytical solution in aquifers under harmonic temperature boundary conditions and constant vertical flow. Later, Bredehoeft and Papadopulos [10] simplified Suzuki's solution for the case of constant temperature on the soil surface.

Most applications of this relationship have led to quantifying recharge [11–14], for instance, the flow interaction between a river and the aquifer [15–17] and applications in shallow aquifers [18–20]. Duque et al. [21] estimated the upward vertical velocity in a groundwater–surface water exchange scenario (a coastal lagoon) using the analytical solution of Bredehoeft and Papadopulos [10]. Woodbury and Smith [22] solved a 3D problem that contains a buoyancy term in the water flow equation but no dispersion term in the heat flow equation. Regarding the methodology of the inverse problem, few examples can be found in the scientific literature of groundwater flow estimated at different depths from temperature data [23].

The experimental temperature data provided in this work, mainly consisting of temperature–depth profile time series (TDPTS hereafter), will be used for the first time to quantify groundwater vertical flow in the coastal aquifer of Agua Amarga (southeastern Spain). It will be implemented by applying the inverse protocol to a numerically simulated 1D problem. In previous studies (2008–2010 period), TDPTS measurements were used, but to qualitatively describe groundwater flow components. An upward movement below the

salt marsh located on the aquifer surface was found in the nearby P-8. The analysis was supported by a fluid flow and solute transport model and hydrochemical data [24]. As an alternative proposal not found in the literature, averaged vertical flow velocity is calculated from the TDPTS data for different depth intervals, so the assumed decrease of velocity toward the surface can be reproduced. Despite its limited extension, the natural scenario selected for this purpose, a small endorheic basin of approximately 1700 ha (Figure 1), has been the framework for successive anthropogenic interventions for the last hundred years, involving the saltwork industry, groundwater abstractions, and seawater recharge. This is a case study that has eventually entailed an experience of coastal aquifer management, demonstrating the positive effect of the measures implemented on the piezometry and salinity decontamination [25]. After more than a decade of aquifer recharge, its effect on flow patterns under the irrigation area is a question of interest that we deal with in this study.

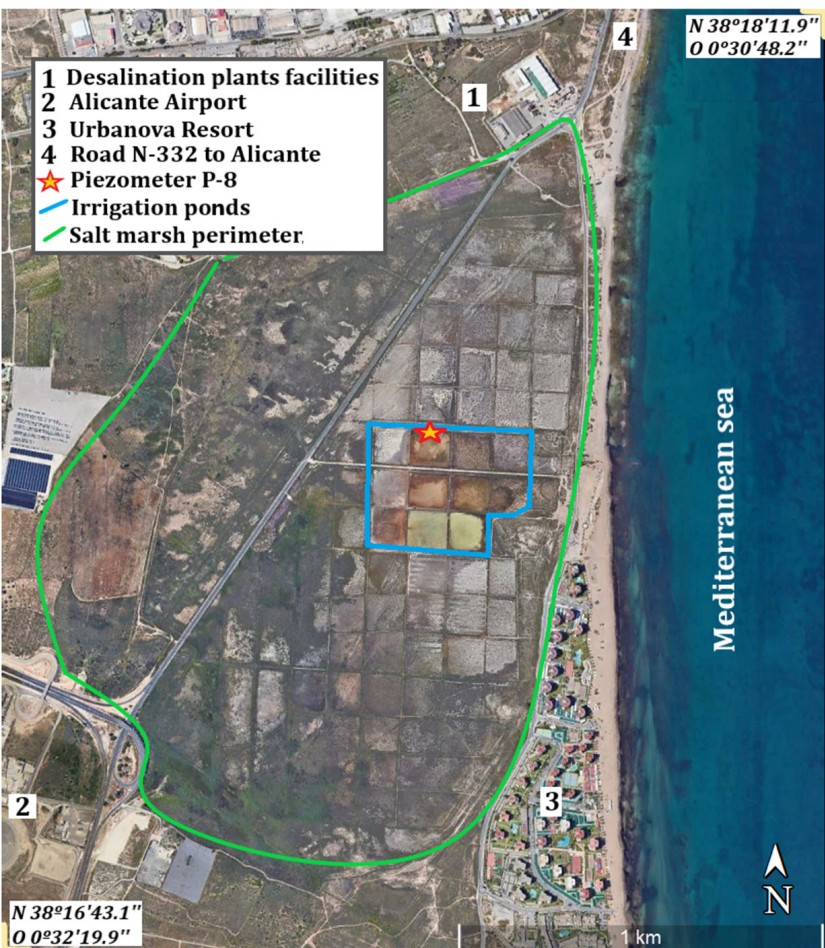

**Figure 1.** The Agua Amarga salt marsh and the seawater irrigation area. Geographic coordinates are in the corner (Base image from Google Earth [California, USA], Landsat Copernicus [USA], and Maxar Technologies [Colorado, USA]).

The information related to groundwater temperature, volumes of seawater used in the recharge, surface air, and seawater temperature and rainfall was gathered thanks to several collaboration agreements during the period 2009–2022 between the Technical University of Cartagena and the Mancomunidad de los Canales del Taibilla (MCT, hereafter), the administrative body responsible for the exploitation of desalination plants (Ministry of Ecological Transition and Demographic Challenge of the Spanish Government). These data, after processing, constitute the experimental input in the inverse protocol that is addressed in two separate ways. The first (P-I) encompasses the whole monitoring period for selected

months (July and August), comparing the TDPTS between years with and without recharge (2009–2015 and 2016–2022, respectively). Constant monthly average temperature boundary conditions are set on the top and bottom layers. The second (P-II) consists of understanding the differences in TDPTS by comparing a year with negligible recharge (2011) with another with paramount recharge (2022). In this problem, sinusoidal and constant temperature boundary conditions are set, respectively, on the top and bottom layers of the model with a monthly mean value.

## 2. Study Area

During the period 1925–1975, a company extracting salt from seawater operated in the Agua Amarga salt marsh [26], taking advantage of its lagoon geomorphology (flat surface). Because of this, the surface was compartmentalized into ponds, and salt deposits formed in them [27]. After this activity concluded, natural evolution led to the development of a salt marsh, which is nowadays under special environmental protection and included in the Valencia Community wetlands catalogue. When the Alicante Desalination Plants I and II (AD-I and AD-II hereafter) started abstracting groundwater from the aquifer along the coastline in 2008, it was necessary to compensate the piezometric depletion below the salt marsh caused by a permanent depression cone located on the coastline (depths ranging from −5 to −13 m b.s.l. [24]). The MCT decided to promote pilot-experience seawater recharging in some ponds of the salt marsh. Since then, monthly surveys have been carried out by the Technical University of Cartagena to measure and control the piezometry, salinity, temperature, soil properties, and vegetation growth.

The groundwater abstraction system for AD-II consists of a 1 km-long tunnel with 104 inclined drains located 10 m deep and separated 50 m from the coastline. An additional supply of seawater is guaranteed by 11 horizontal directional drilling wells under the seabed. In the case of AD-I, 18 vertical pumping wells were installed along the coastline in the northern part of the salt marsh. A complete explanation of the intake system was made by Rodríguez-Estrella and Pulido-Bosch [28]. The hydrogeological connection between the upper aquifer and the sea, described by Alhama [29], guarantees seawater flow to the abstraction system (approximately 87–98% seawater). The aquifer consists of Tyrrhenian sandstone, with an average thickness of 50 m, intercalated with continental silt, sands, and gravel deposits of Pliocene-Pleistocene, all of which are covered with surface soils made of clayed mud with gypsum, salt, and fine sand 1 to 5 m thick. The existence of brine (reaching 400 mS/cm values in 2008 field surveys), piezometric lowering below the salt marsh due to groundwater abstraction, the special care required for surface ecosystems, and the accessibility to seawater led to the decision to restore the aquifer by recharging it from the surface using seawater. A description of the aquifer recharge system, the effects on the aquifer, and its economic impact are described in Navarro and Sánchez-Lizaso [30] and Alhama et al. [31]. From an environmental point of view, the seawater recharge measure throughout the period 2010–2022 has proved to be effective and sustainable for aquifer desalination, piezometric restoration, and vegetation growth since they are all compatible with natural resource exploitation, as concluded in the most recent work on the area [25].

Other studies carried out in the last decade have been aimed at understanding the dynamics of the aquifer from the experimental data gathered while exploring and monitoring the aquifer, as in this work. For instance, to determine the hydraulic conductivity distribution in the aquifer and qualitatively describe flow patterns driven by salinity concentration gradients, Alhama [32] used piezometric and salinity data to calibrate a numerical MODFLOW model.

## 3. Methodology

Simulations of the scenarios have been carried out using numerical methods so that real data can be reproduced. This has involved a trial-and-error process consisting of varying the boundary conditions in consecutive simulations (calibration process). This process has allowed us to quantify vertical velocities.

The inverse method requires suitable and organized information to calibrate the model and set boundary conditions. We have also included the procedure applied to experimental data processing.

### 3.1. Experimental Data Introduction and Processing

Data related to rainfall and temperature were gathered from the Alicante airport weather station in El Altet, placed 2 km southwest from the salt marsh. The information is available to the public on the Internet [33]. Mean monthly air temperature is considered for the top boundary condition when recharge is neglected, and seawater temperature is set as a boundary condition in months when the recharge is active on the salt marsh surface. This is because the water for irrigation comes from the catchment system of DA-II (98% seawater [29]). The heat exchange between the seawater deposited over the salt marsh (artificial recharge) and the air before infiltration has been ignored because of the differences of specific heat between water and air (4 times lower), the similitude of monthly temperature time series throughout the period for both (Figure 2), and the fact that the permanent sheet of water remains only for a maximum of one day after the irrigation has stopped, which is little time to interact with the air.

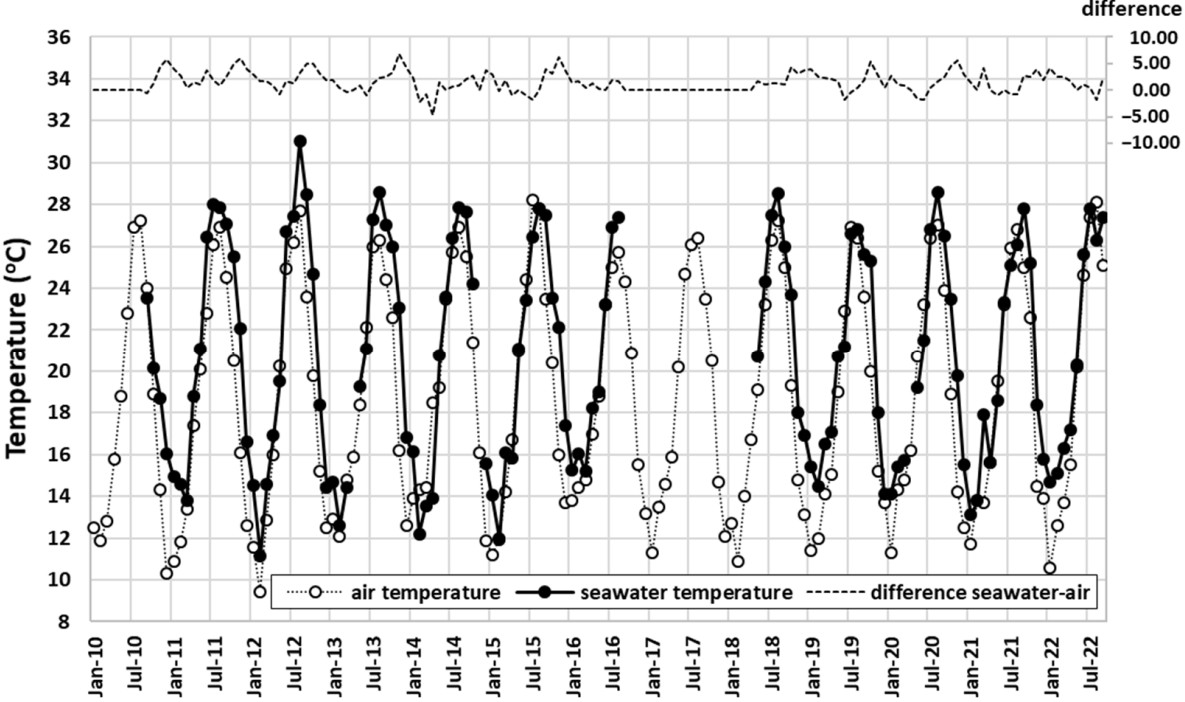

**Figure 2.** Average monthly temperature of air and seawater and differences (°C).

The monthly volumes of seawater involved in the recharge for the whole period have been provided by MCT (Figure 3). Only those affecting the area where piezometer P-8 is included (Figure 1) have been considered in this work. To compare this volume with natural recharge (rainfall), an interval of infiltration (mm) has been calculated each month. For a minimum irrigation value, the volume has been divided by the total area of infiltration (8 ponds or evaporation pans reaching an area of approximately 93,000 m$^2$), whereas for a maximum irrigation value, only the area where a permanent sheet was observed is considered. Even though the irrigation pipe system surrounds the 8 ponds, not all of them were activated simultaneously, and seawater does not frequently cover the whole area.

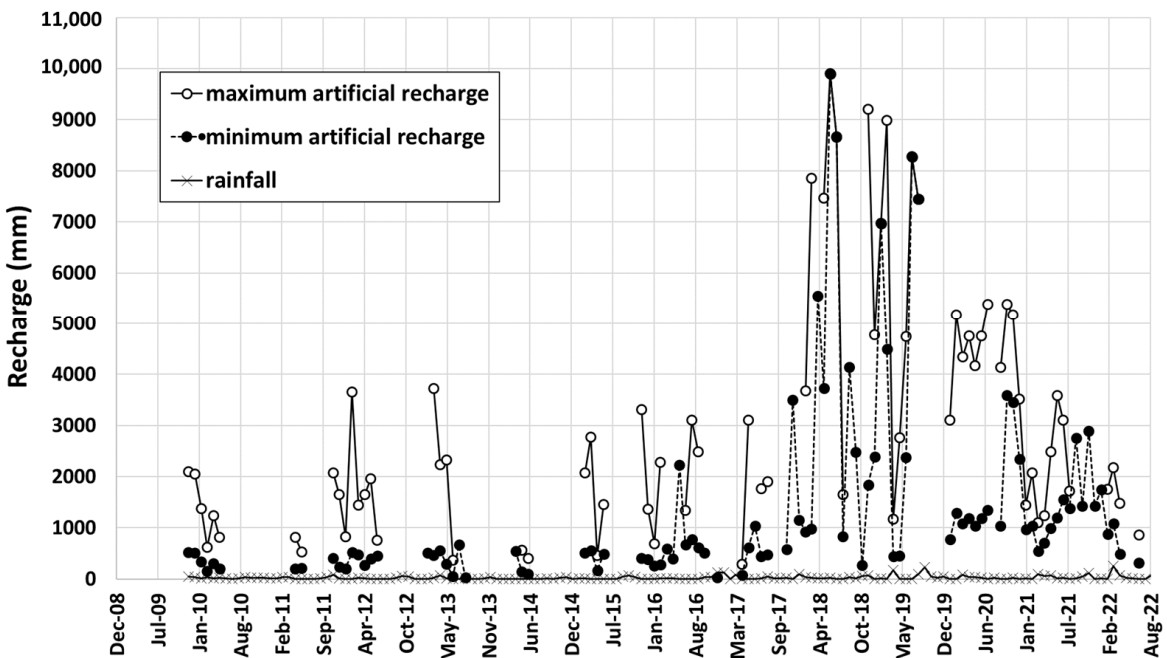

**Figure 3.** Maximum and minimum recharge on the salt marsh together with rainfall.

As shown in Figure 3, natural recharge from the surface can be negligible compared to the volume coming from artificial irrigation. Artificial recharge has not been constant throughout the period, with years when it was negligible (2011, 2014) and others when it was uninterrupted (2020).

Since 2010, the groundwater temperatures in piezometer P-8 have been measured monthly. During the data acquisition, the probe (Heron Conductivity Plus model) is slowly introduced into the piezometer, and once the groundwater surface is reached, the stabilized temperature value is recorded at every meter depth until the bottom of the piezometer is reached (16 m). The information represents the period from January 2010 to October 2022, except for the interval between September 2016 and April 2018. Average monthly groundwater temperatures for the whole period at every meter depth are shown in Figure 4. The accuracy of the probe, calibrated in regular laboratory testing, allows the temperature to be depicted in decimals.

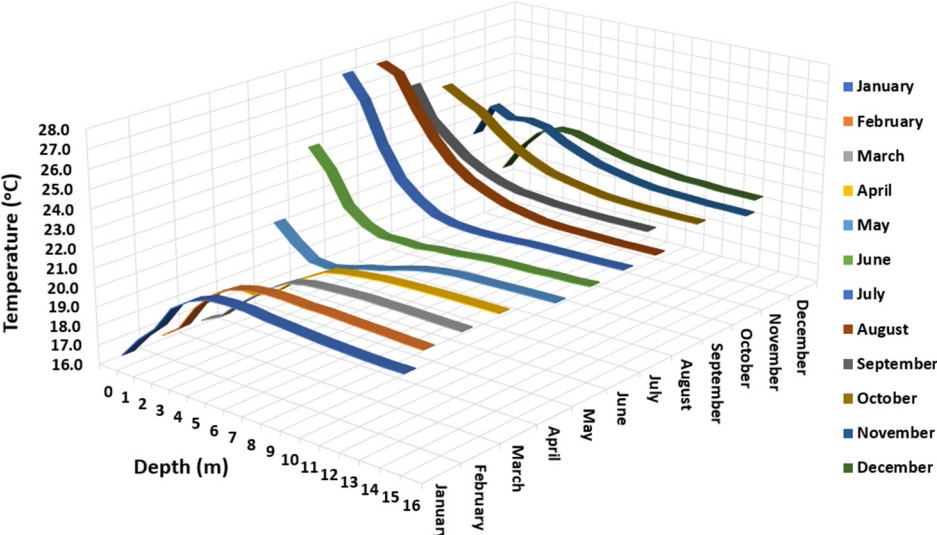

**Figure 4.** Average TDPTS for the whole period in piezometer P-8.

The study of the groundwater flow components below the salt marsh is addressed by assessing the TDPTS from two different points of view. In P-I, July and August have been selected as months that include years with and without recharge (2010–2015 and 2016–2022, respectively). The total volume of seawater used for irrigation is depicted in Figure 5, and the average TDPTS for those periods are shown in Figure 6.

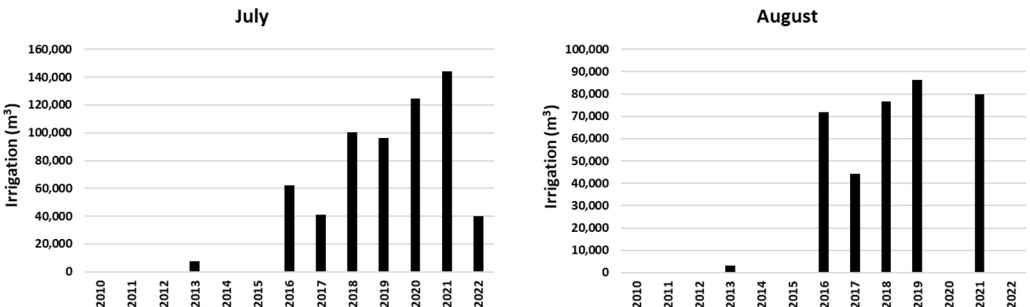

**Figure 5.** Total volume of seawater used for irrigation in July (**left**) and August (**right**) for the period 2010–2022.

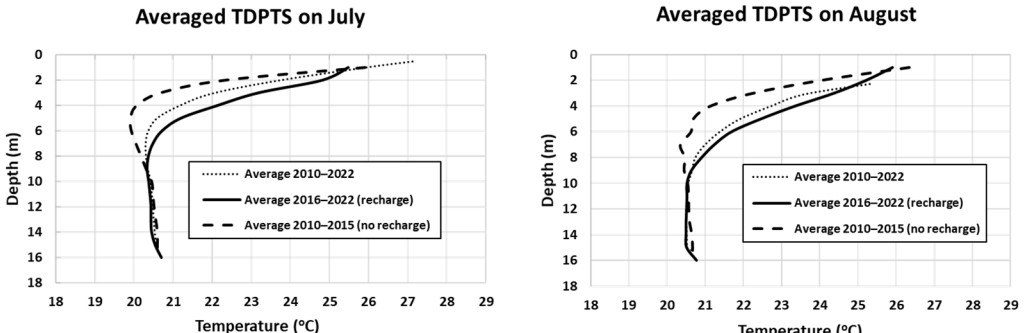

**Figure 6.** TDPTS averaged for years with and without recharge (2010–2015 and 2016–2022, respectively) in the months of July (**left**) and August (**right**).

P-II compares a year when the recharge is negligible (2011) with another with maximum and continuous recharge (2020). The influence of TI is insignificant since the recharge in previous years can be disregarded before 2011, and it is high before 2020 (2016–2019 interval, Figure 3). The TDPTS for piezometer P-8 in 2011 and 2020 are depicted in Figure 7.

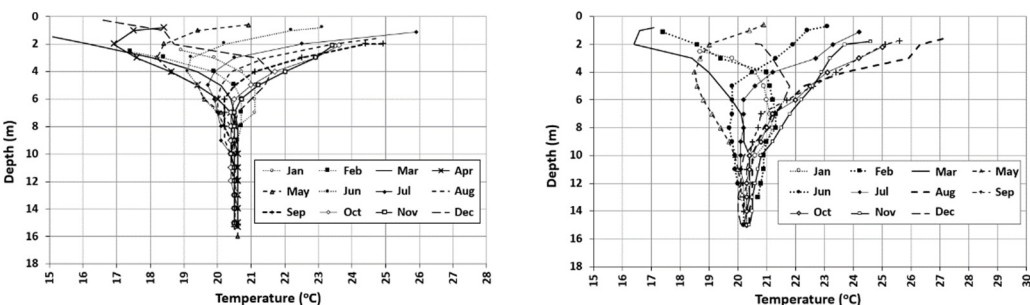

**Figure 7.** TDPTS for piezometer P-8 in 2011 (no artificial recharge, **left**) and 2020 (intensive artificial recharge, **right**).

As can be observed, artificial recharge drags heat or cold from the surface, enlarging the interval of temperature variation at every depth.

### 3.2. The Inverse Problem Protocol

To deal with the application of the inverse protocol in this work, consecutive stages are required. First, as addressed in the previous section, organized and suitable experi-

mental information is needed. Second, a numerical method to simulate the scenarios for specific and variable thermal and mechanical boundary conditions must be developed. The definition of a function helps compare the numerical results (simulated) with real data so that the unknown variable can be assessed (vertical velocity).

As the numerical model is described in the next section, we focus here on the functional and calibration process. The functional process is defined for four different depths (z) since vertical velocity changes with depth. The following formula is applied to calculate it:

$$\varphi_z = [T_z - T_{z\ simulated}]^2 \tag{1}$$

The functional variations are represented graphically to select the optimal one for each depth. For the first simulation, an initial (low) groundwater velocity, $v_{z,o,1}$, is applied. Successively changing velocity, for example $v_{z,o,1} + \Delta v_z$, $v_{z,o,1} + 2\Delta v_z$, $v_{z,o,1} + 3\Delta v_z \ldots$ , the function in the successive simulations is obtained until its value is small enough ($\varphi_z \approx 0$). Beyond this point, the functional value increases again.

## 4. Numerical Model

The inverse method requires numerical simulation at a certain step to assess how far our physical model is from reality (calibration). In this section, we first deal with the definition of the physical and mathematical models. After some brief notes about the network method, the network model used to numerically solve the problem is then described. Finally, we summarize all the input data to be set in every scenario studied (6 altogether), according to the descriptions given to P-I and P-II.

The geometric scheme and the thermal boundary conditions of the problem are shown in Figure 8. The origin of the coordinates is on the surface, increasing the z coordinate in depth. As for the TDPTS, the model only considers the saturated area (below the phreatic level).

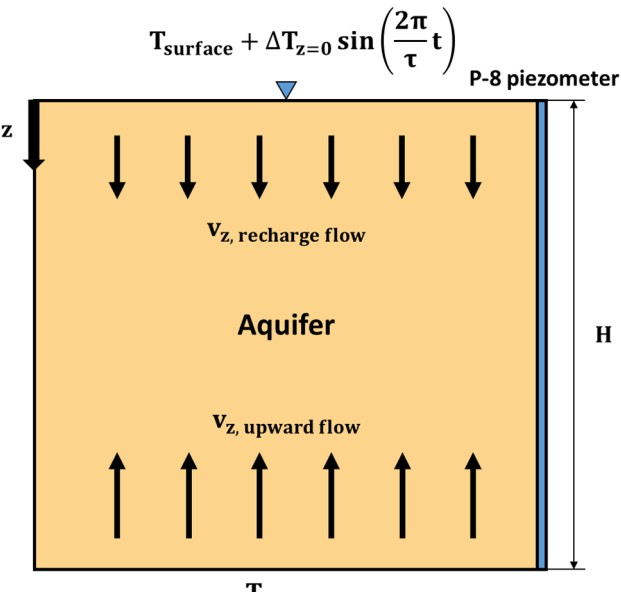

**Figure 8.** Physical scheme and boundary conditions of the 1D problem.

The governing equation and boundary conditions are as follows:

$$k_m \left( \frac{\partial^2 T}{\partial z^2} \right) - \rho_{e,w} c_{e,w} v_z \frac{\partial T}{\partial z} - \rho_e c_e \frac{\partial T}{\partial t} = 0 \tag{2}$$

$$v(z, t) = v_z \tag{3}$$

$$T_{z=0} = T_{surface} \tag{4}$$

$$T_{(z=0,t)} = T_{surface} + \Delta T_{z=0} \sin\left(\frac{2\pi}{\tau}t\right) \tag{5}$$

$$T_{z=H} = T_{bottom} \tag{6}$$

$$T_{(z,t=0)} = T_{ini} \tag{7}$$

The governing Equation (2) is that of the simultaneous flow of heat and fluid in homogeneous, isotropic, and rectangular domains [34]. This equation represents the local balance of heat fluxes: diffusion $\left(k_m \frac{\partial^2 T}{\partial z^2}\right)$, advection $\left(\rho_{e,w} c_{e,w} v_{z,o} \frac{\partial T}{\partial z}\right)$, and storage $\left(\rho_e c_e \frac{\partial T}{\partial t}\right)$. The groundwater flow in the porous media is vertical and one-dimensional (upward), and the velocity has a constant value for a specific depth (3) in each simulation. The aquifer surface is under a Dirichlet-type boundary condition of constant temperature (4) for scenarios 1 to 4 (P-I). The temperature undergoes a sinusoidal harmonic variation (5) for scenarios 5 to 6. The bottom of the domain is under constant temperature averaged from the TDPTS in all circumstances. Finally, the initial condition, although it has no influence on the steady-state temperature solution, determines the value of transient temperature–depth profiles $T_{(z,t)}$.

There are many numerical tools to solve heat transport equation in porous media, for instance, MODFLOW with the package MT3DMS [35,36] or VS2DHI [37]. Some of these numerical tools have been integrated into commercial software (interface) for research and professional purposes, for example, GMS [38,39]. In this work, we have developed a numerical model based on the analogy of governing equations to those ruling electric circuits. Its design follows the rules of the network simulation method [40], a tool that has demonstrated its computational efficiency and reliability in other engineering problems of similar complexity [41]. This method has also shown its possibilities in the solution of benchmark problems related to transport in porous media, such as Bénard [42], Yusa [43], and Henry [44].

To create a network model, the first step is to reticulate the space into elementary cells and define the nodes. This method starts from the finite-difference differential equation(s) that result from the spatial discretization of the mathematical model. Each term of the equation is implemented like a current in a branch whose constitutive (voltage–current) equation is that of the term. Each branch converges in a common node where all the currents are balanced. Time is a continuous variable in the simulation. The spatially discretized equations of the mathematical model and the network model are formally equivalent. The complete model, once the connection between volume elements has been made through ideal electric contacts, is run in the free software Ngspice [45] or Pspice [46]. The solution, thanks to the powerful mathematical algorithms implemented, is nearly the exact solution of the model. Any errors are linked to the choice of the grid mesh size. In our case, the problem is 100 cells deep.

To obtain the equations of the elements, it is first necessary to spatially discretize the governing equation $k_m \left(\frac{\partial^2 T}{\partial z^2}\right) - \rho_{e,w} c_{e,w} v_z \frac{\partial T}{\partial z} - \rho_e c_e$ as follows:

$$k_m \left[\frac{1}{z}\left(\left.\frac{\Delta T}{\Delta z}\right|_{z^+} - \left.\frac{\Delta T}{\Delta z}\right|_{z^-}\right)\right] - \rho_{e,w} c_{e,w} v_z \frac{\Delta T}{\Delta z} - (\rho_e c_e)\frac{dT}{dt} = 0 \tag{8}$$

Secondly, the equation should be expressed in finite differences:

$$\frac{(T)_{i+\frac{\Delta z}{2},j} - (T)_{i,j}}{\frac{(\Delta z)^2}{2k_m}} + \frac{(T)_{i,j} - (T)_{i-\frac{\Delta z}{2},j}}{\frac{(\Delta z)^2}{2k_m}} - \rho_{e,w} c_{e,w} v_z \frac{(T)_{i+\frac{\Delta z}{2},j} - (T)_{i+\frac{\Delta z}{2},j}}{\Delta z} - (\rho_e c_e)\frac{dT_{i,j}}{dt} = 0 \tag{9}$$

From the governing equation in finite differences, the equations of each of the elements are obtained:

$$R_{cu,i} = R_{cd,i} = \frac{(\Delta z)^2}{2 \cdot k_m} \tag{10}$$

$$G_{cu,i} = \rho_{e,w} c_{e,w} \frac{v_z V(i)}{\Delta z} \tag{11}$$

$$G_{cd,i} = \rho_{e,w} c_{e,w} \frac{v_z V(i+1)}{\Delta z} \tag{12}$$

$$J_C = \rho_e c_e \frac{dT(t)}{dt} \tag{13}$$

In addition, the initial temperature of the cells must be set in the capacitors. In the cells located on the soil surface and in the cell at the bottom of the domain, batteries have been connected to generate a voltage of constant value. In the network model, the following analogy between variables stands for:

Heat flow ≡ Current intensity

Temperature ≡ Electric potential

Therefore, setting a constant electric potential on the surface and at the bottom of the aquifer is equivalent to setting a constant temperature at both boundaries. For the scenario in which the temperature varies sinusoidally on the surface, a cell is also connected. This indicates that its voltage varies sinusoidally, defining the mean value, amplitude, and period.

To study the thermal inertia and modify the value of velocity in the same model (studying the transient), it is necessary to create a new node and connect to it an auxiliary resistor and a time-dependent voltage source. The resistance will have a constant value, while the voltage from the auxiliary voltage source will change its value in the same way as the groundwater velocity.

Since the water velocity is not of constant value throughout the simulation in this transient model, the new expression of generators is modified as follows:

$$G_{cu,i} = \rho_{e,w} c_{e,w} \frac{V(102)V(i)}{\Delta z} \tag{14}$$

$$G_{cd,i} = \rho_{e,w} c_{e,w} \frac{V(102)V(i+1)}{\Delta z} \tag{15}$$

As mentioned before, and bearing in mind the amount of available information, two problems (P-I and P-II) are addressed in this work. In P-I, July and August are the months selected to study the TDPTS for the whole period. Since we can identify two periods, one with negligible recharge (2013–2015) and the other with significant recharge (2016–2022), P-I can be divided into four scenarios (see Figure 5). The values of the boundary conditions and parameters are summarized in Table 1.

**Table 1.** Boundary conditions and parameters defined in scenarios 1–4 of P-I.

| Scenario | Time Interval | $k_m$ (cal $°C^{-1}$ $s^{-1}$ $m^{-1}$) | $\rho_e c_e$ (cal $°C^{-1}$ $m^{-3}$) | $T_{surface}$ (°C) | $T_{bottom}$ (°C) |
|---|---|---|---|---|---|
| 1 | July 2010–2015 | 0.45 | $0.75 \times 10^{-6}$ | Average air temperature 2010–2015: 26.5 | Average temperature at 16 m depth in the piezometer 2010–2015: 20.5 |
| 2 | July 2016–2022 | 0.45 | $0.75 \times 10^{-6}$ | Average seawater temperature 2016–2022: 26.8 | Average temperature at 16 m depth in the piezometer 2016–2022: 20.5 |
| 3 | August 2010–2015 | 0.45 | $0.75 \times 10^{-6}$ | Average air temperature 2010–2015: 27.1 | Average temperature at 16 m depth in the piezometer 2010–2015: 20.6 |
| 4 | August 2016–2022 | 0.45 | $0.75 \times 10^{-6}$ | Average seawater temperature 2016–2022: 27.3 | Average temperature at 16 m depth in the piezometer 2016–2022: 20.5 |

In the case of P-II, we study the TDPTS in 2011 and 2020. The first is not influenced by irrigation during the previous years, whereas the second is preceded by a period with significant irrigation (see Figure 3). We define two new scenarios, one for 2011 (scenario 5) and the other for 2020 (scenario 6). Table 2 summarizes the boundary conditions and parameters used in the numerical modeling.

**Table 2.** Boundary conditions and parameters defined in scenarios 5–6 of P-II.

| Scenario | Time Interval | $k_m$ (cal $°C^{-1}$ $s^{-1}$ $m^{-1}$) | $\rho_e c_e$ (cal $°C^{-1}$ $m^{-3}$) | $T_{surface}$ (°C) | $T_{bottom}$ (°C) |
|---|---|---|---|---|---|
| 5 | 2011 | 0.45 | $0.75 \times 10^{-6}$ | Sinusoidal function based on monthly air temperature 2011 | Constant value calculated from mean temperature at 16 m depth 2011: 20.6 |
| 6 | 2020 | 0.45 | $0.75 \times 10^{-6}$ | Sinusoidal function based on monthly air temperature 2020 | Constant value calculated from mean temperature at 16 m depth 2020: 20.3 |

A vertical downward boundary condition has not been considered in scenarios 2, 4, and 6 since the water flowing into the aquifer from the surface is horizontally diverted close to the surface. However, the effect of this recharge is implied in the lessening of upward velocity, as will be discussed later.

## 5. Inverse Problem Applied to Vertical Flow Analysis

Once the experimental data has been organized and the models defined, the protocol of the inverse problem can be applied. For P-I, the model has been defined according to the data included in Table 1. The variations of the function for each scenario at the selected depths are depicted in Figure 9.

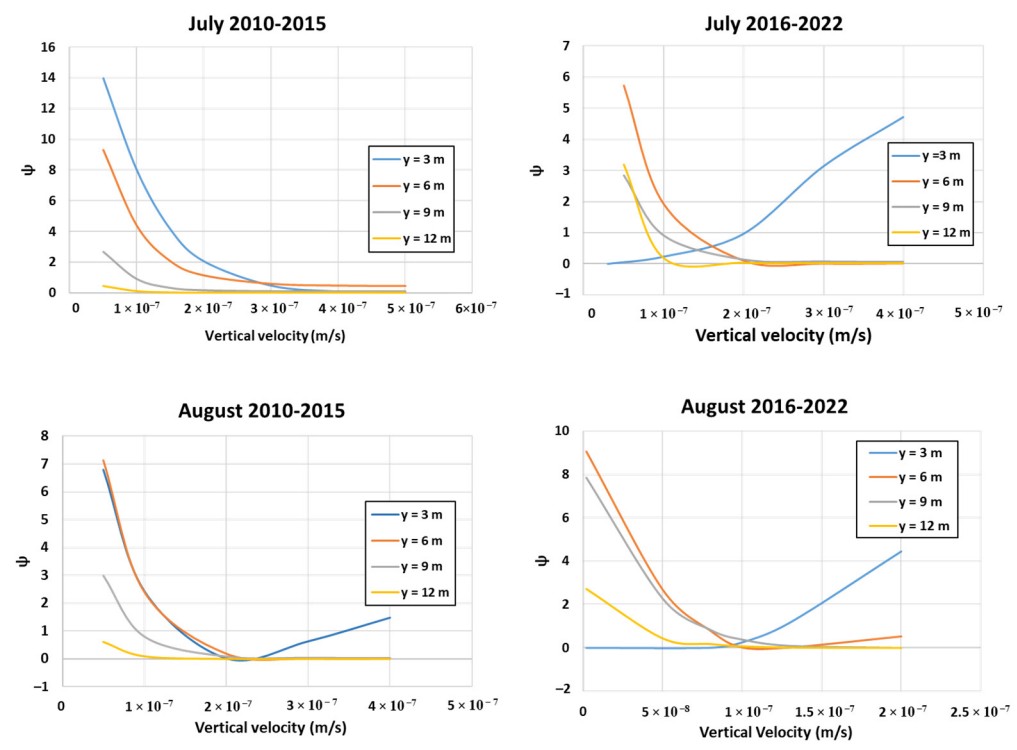

**Figure 9.** Influence of velocity on function variations ($\Psi$) for each scenario at the selected depths.

Accordingly, the minimum value of the function defines the optimal velocity. Table 3 summarizes the selected values.

**Table 3.** Selected velocities for every depth according to the protocol in P-I. The symbol [*] refers to the results of limited reliability.

| Scenario | Velocity at 3 m Depth (m/s) | Velocity at 6 m Depth (m/s) | Velocity at 9 m Depth (m/s) | Velocity at 12 m Depth (m/s) |
|---|---|---|---|---|
| 1 | $7 \times 10^{-7}$ | $7 \times 10^{-7}$ | $7 \times 10^{-7}$ | $7 \times 10^{-7}$ |
| 2 | $3 \times 10^{-8}$ * | $4 \times 10^{-7}$ | $4 \times 10^{-7}$ | $4 \times 10^{-7}$ |
| 3 | $2 \times 10^{-7}$ | $4 \times 10^{-7}$ | $4 \times 10^{-7}$ | $4 \times 10^{-7}$ |
| 4 | $2 \times 10^{-9}$ * | $1 \times 10^{-7}$ | $2 \times 10^{-7}$ | $2 \times 10^{-7}$ |

In P-II, the model has been defined according to the data included in Table 2. As upward vertical velocity increases with depth, the value fitting has also been carried out at different depths. The calibration has been applied only to the months considered close to steady since this protocol has not been designed to solve transient states. Table 4 summarizes the selected values.

**Table 4.** Selected velocities for every depth according to the protocol in P-II for scenarios 5 (2011) and 6 (2020). The symbol [*] refers to results of limited reliability.

| Month (Year) | Velocity at 3 m Depth (m/s) | Velocity at 6 m Depth (m/s) | Velocity at 9 m Depth (m/s) | Velocity at 12 m Depth (m/s) |
|---|---|---|---|---|
| January (2011) | $4.0 \times 10^{-7}$ | $6.0 \times 10^{-7}$ | $6.0 \times 10^{-7}$ | $6.0 \times 10^{-7}$ |
| February (2011) | $1.5 \times 10^{-7}$ | $7.5 \times 10^{-7}$ | $7.5 \times 10^{-7}$ | $7.5 \times 10^{-7}$ |
| March (2011) | $1.0 \times 10^{-7}$ | $2.5 \times 10^{-7}$ | $5.0 \times 10^{-7}$ | $5.0 \times 10^{-7}$ |
| August (2011) | $2.5 \times 10^{-7}$ | $7.5 \times 10^{-7}$ | $7.5 \times 10^{-7}$ | $7.5 \times 10^{-7}$ |
| September (2011) | $0.0$ * | $7.0 \times 10^{-7}$ | $7.0 \times 10^{-7}$ | $7.0 \times 10^{-7}$ |
| October (2011) | $0.0$ * | $7.0 \times 10^{-7}$ | $7.0 \times 10^{-7}$ | $7.0 \times 10^{-7}$ |
| November (2011) | $1.0 \times 10^{-8}$ * | $2.0 \times 10^{-7}$ | $5.0 \times 10^{-7}$ | $5.0 \times 10^{-7}$ |
| January (2020) | $4.0 \times 10^{-7}$ | $5.0 \times 10^{-7}$ | $5.0 \times 10^{-7}$ | $5.0 \times 10^{-7}$ |
| February (2020) | $1.5 \times 10^{-7}$ | $7.0 \times 10^{-7}$ | $7.0 \times 10^{-7}$ | $7.0 \times 10^{-7}$ |
| March (2020) | $1.5 \times 10^{-7}$ | $1.5 \times 10^{-7}$ | $1.5 \times 10^{-7}$ | $3.5 \times 10^{-7}$ |
| June (2020) | $5.0 \times 10^{-8}$ * | $6.0 \times 10^{-7}$ | $6.0 \times 10^{-7}$ | $6.0 \times 10^{-7}$ |
| July (2020) | $-3.0 \times 10^{-8}$ * | $7.0 \times 10^{-7}$ | $7.0 \times 10^{-7}$ | $7.0 \times 10^{-7}$ |
| August (2020) | $-2.0 \times 10^{-7}$ | $8.0 \times 10^{-8}$ | $2.0 \times 10^{-7}$ | $2.0 \times 10^{-7}$ |
| September (2020) | $4.0 \times 10^{-9}$ * | $8.0 \times 10^{-8}$ | $1.5 \times 10^{-7}$ | $4.0 \times 10^{-7}$ |
| October (2020) | $4.5 \times 10^{-8}$ * | $8.0 \times 10^{-8}$ | $1.0 \times 10^{-7}$ | $4.0 \times 10^{-7}$ |
| November (2020) | $3.5 \times 10^{-8}$ * | $3.5 \times 10^{-8}$ | $6.0 \times 10^{-8}$ | $1.0 \times 10^{-7}$ |

As expected, the calibration illustrates that upward velocity decreases near the surface, where horizontal component values increase to compensate groundwater mass balance. Moreover, comparing any month in 2011 and 2020, we can observe that the effect of irrigation causes an additional lowering of upward velocities. When no recharge is implemented (scenarios 1, 3, and 5), vertical upward flow occurs even close to the phreatic level, although it is negligible. However, when irrigation takes place (scenarios 2, 4, and 6), zero vertical upward velocity is located at some depth below the phreatic level, and vertical downward flow occurs above this position to adjust the irrigation (see July and August 2020).

## 6. Discussion

To make the results consistent, the starting point of the discussion responds to the fact that we have omitted horizontal flow from the 1D model analysis. Water flowing into the aquifer from the top or bottom boundaries is leaving laterally. If we consider a column in which no vertical flow boundary condition is set and temperature boundary conditions are those of, for example, scenario 5, the simulation generates a TDPTS like those depicted in Figure 10.

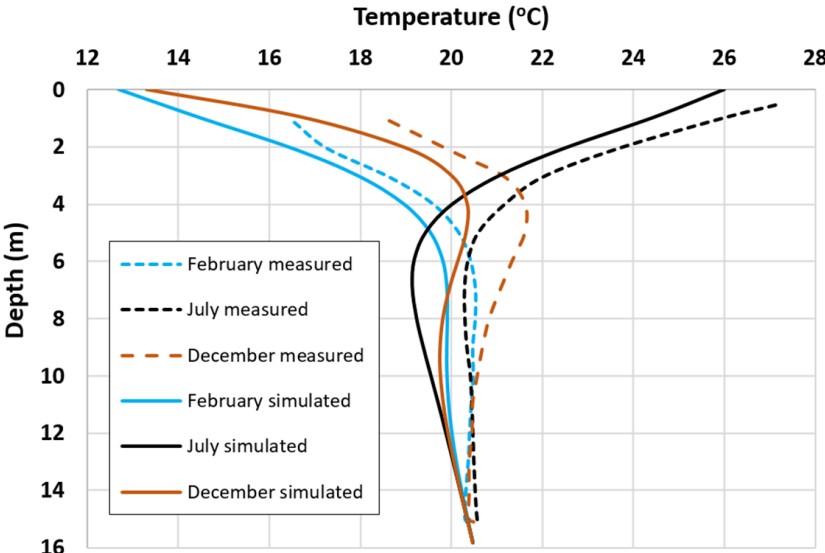

**Figure 10.** TDPTS simulated with no groundwater flow vs. TDPTS measured.

As the temperature boundary conditions are similar, the comparison of the TDPTS shows that heat transfer from the bottom must be taking place. Simulated TDPTS are located to the left of the real ones, and these remain vertical at a certain distance from the bottom (below 10 m depth) due to advection. The TDPTS is independent of the value assigned to the horizontal flow velocity and a function of thermal conductivity and diffusivity. Considering them as constant and comparing the TDPTS in Figure 10 month to month, the differences between them can only be fitted by setting vertical upward flow boundary conditions.

The existence of upward vertical flow components in coastal aquifers has been widely discussed and proved. Leaving aside the geological aspect that conditions flow, homogeneous aquifers deal with buoyancy effects that control flow direction, especially near the intrusion wedge or transition zone. The values of upward flow velocity obtained in this work are similar, or fit the same order of magnitude, as those obtained in Ringkøbing Fjord (North Sea, Denmark) by Duque et al. [21] and Tirado-Conde [5]. The first values range from $3.4 \times 10^{-7}$ to $7.3 \times 10^{-7}$ m/s by applying Bredehoeft and Papadopulos' solution [10], and the second, using the seepage meters procedure, has an interval of $2.3 \times 10^{-9}$–$1.2 \times 10^{-6}$ m/s. Employing Bredehoeft and Papadopulos' solution, we obtain an average of $3.5 \times 10^{-7}$ m/s. Other scenarios in which temperature data were used to estimate vertical velocities had similar results. Lapham [1] studied upward flow in a river–groundwater interaction scenario in two different locations in the US (Hardwick and New Braintree). The values ranged from $3.5 \times 10^{-8}$ m/s to $7 \times 10^{-7}$ m/s. According to Stallman [34], percolation rates of $2.3 \times 10^{-7}$ m/s can be detected with profiles resulting from diurnal temperature fluctuations. This rate can go down to $0.35 \times 10^{-7}$ m/s in low conductivity soils with a wide range of temperature variations and carefully taken measurements. Considering annual temperature fluctuations, detection can reach values of $0.11 \times 10^{-7}$ m/s.

As mentioned in previous sections, TI influences the TDPTS when temperature boundary conditions or vertical flows change. The effect of TI has been assessed by using the model described in Section 3. Observing temperatures in the middle of the simulated piezometer (8 m depth), the transient time spent to change from two different steady-state temperatures as a consequence of vertical variation has been measured. Table 5 summarizes some results.

**Table 5.** Transient time simulated as a result of a change in vertical velocity.

| $V_{z,1}$ (m/s) | $V_{z,2}$ (m/s) | Transient Time (days) |
|---|---|---|
| $0.0 \times 10^{-7}$ | $7.00 \times 10^{-7}$ | 111 |
| $1.00 \times 10^{-7}$ | $7.00 \times 10^{-7}$ | 104 |
| $2.00 \times 10^{-7}$ | $7.00 \times 10^{-7}$ | 82 |
| $4.00 \times 10^{-7}$ | $7.00 \times 10^{-7}$ | 65 |

According to the data in Table 5, the effect of TI can be considered negligible in P-II since before 2011 and 2020, irrigation was almost null (2010) or continuous (2016–2019). The steady-state conditions assumption can then be considered suitable. In the case of P-I, a transient time ranging from 4 to 2 months would be necessary to reach a steady state. According to Figure 3 and examining the preceding months of July and August, before 2015, irrigation was null or negligible, and, for the period 2016–2020, the artificial recharge was continuous. The assumption of a steady state in the approach of P-I is not strictly rigorous, but it is a reasonable hypothesis.

As the numeral model used does not contemplate the effects of buoyancy on flow, the validity of the results previously achieved should be discussed. The effect of variable density on flow has been addressed firstly by considering the salinity changes and then the temperature gradients. In the first place, the possible density changes due to salinity variations have been addressed by taking into account the salinity-depth profiles registered in piezometer P-8. These profiles were published in Figure 5 in the work of Alhama et al. [25]. By observing these profiles, we can affirm that since 2016, the ECgs (mS/cm) has been constant at all depths. Therefore, there is no vertical flow due to density variation produced by differences in salinity.

Regarding the flow induced by density variations caused by temperature changes, the equation reflected in Cánovas et al. [42] has been used to calculate $v_{bouyancy}$:

$$v_{bouyancy} = \frac{\rho_{e,w} k \beta g}{\mu} \Delta T_{interval} \tag{16}$$

where

$$k = \frac{\mu K_{Darcy}}{\rho_{e,w}} \tag{17}$$

Thus, Equation (15) is rewritten as follows:

$$v_{bouyancy} = K_{Darcy} \beta g \Delta T_{interval} \tag{18}$$

We have considered a value of hydraulic conductivity ($K_{Darcy}$) of $8.6 \times 10^{-7}$ and $3.5 \times 10^{-8}$ m/s from the previous study of Alhama [32]. These values are the lowest attributed to the Pliocene–Quaternary materials in the upper aquifer, which limit the vertical flow (transversal to layers arranged horizontally). These are the standard values of $\mu$ ($1 \times 10^{-3}$ kg m$^{-1}$ s$^{-1}$), $\rho_{e,w}$ ($10^3$ kg m$^{-3}$), g (9.81 m s$^{-2}$), and $\beta$ ($[1,2] \times 10^{-4}$ °C$^{-1}$).

Table 6 shows the minimum and maximum velocities related to the advective term for both P1 and P2 (see Tables 3 and 4) and for depth intervals of 0–4.5, 4.5–7.5, 7.5–10.5, and 10.5–16 m. The maximum buoyancy velocity considering convection effects (density changes due to temperature changes) appears in column 5. It has been obtained by combining the values of permeability and $\beta$. We can see how, for a depth of 3 m (0–4.5 m), the estimated advective velocities are of the same order of magnitude as those estimated for convection (buoyancy velocity). As a consequence, some of the results obtained have limited reliability. For the rest of the depth intervals, the maximum convective velocity is 10 times lower or more than the minimum advective velocity, so the buoyancy effects are negligible, and the results of Tables 3 and 4 are reliable at depths of 6, 9, and 12 m.

**Table 6.** Comparison between advection and convection velocities for each depth interval.

| Depth Interval (m) | $\Delta T_{interval}$ (°C) | $v_{advection}$ P1 (m/s) | $v_{advection}$ P2 (m/s) | $v_{bouyancy}$ (m/s) |
|---|---|---|---|---|
| 0–4.5 | 5.75 | $[2 \times 10^{-9}\text{–}7 \times 10^{-7}]$ | $[0\text{–}4 \times 10^{-7}]$ | $9.79 \times 10^{-9}$ |
| 4.5–7.5 | 2.25 | $[1 \times 10^{-7}\text{–}7 \times 10^{-7}]$ | $[3.5 \times 10^{-8}\text{–}7.5 \times 10^{-7}]$ | $3.83 \times 10^{-9}$ |
| 7.5–10.5 | 1.05 | $[2 \times 10^{-7}\text{–}7 \times 10^{-7}]$ | $[6 \times 10^{-8}\text{–}7.5 \times 10^{-7}]$ | $1.79 \times 10^{-9}$ |
| 10.5–16 | 0.35 | $[2 \times 10^{-7}\text{–}7 \times 10^{-7}]$ | $[1 \times 10^{-7}\text{–}7.5 \times 10^{-7}]$ | $5.96 \times 10^{-10}$ |

## 7. Conclusions

Monitoring TDPTS was carried out in the Agua Amarga coastal aquifer during the period 2010–2022, in which groundwater abstractions and artificial irrigation with seawater were combined. This, together with volumes of water for irrigation, air, and seawater temperatures, was used as input data to apply the inverse problem protocol to determine the vertical velocity of groundwater in the vicinity of a piezometer in the irrigation pond located on the salt marsh surface.

The minimum value of a function in the calibration step defined the objective value of vertical velocity for the six scenarios specified in the two problems outlined. The first (P-I) includes the whole monitoring period for July (scenarios 1 and 2) and August (scenarios 3 and 4), comparing the TDPTS between years with and without recharge (2009–2015 and 2016–2022, respectively). The values of vertical upward velocities ranged from $1 \times 10^{-7}$ to $7 \times 10^{-7}$ m/s. The second (P-II) compared the TDPTS from a year in which the recharge was negligible (2011, scenario 5) with another in which the recharge was significant (2022, scenario 5). The results varied between $3.5 \times 10^{-8}$ and $7.5 \times 10^{-7}$ m/s. In all the scenarios, the velocity decreased near the surface (from $4.9 \times 10^{-7}$ m/s averaged at 12 m depth to $2.8 \times 10^{-7}$ m/s averaged at 3 m depth). It was negative (downward flow) at 3 m depth in scenario 6, in which intensive artificial recharge was taking place.

Horizontal and downward components of the velocities were not studied but are assumed to occur near the surface. TI was also studied, obtaining transient times for the TDPTS to reach steady-state conditions in such a way that it can be ensured that the profiles used in the six scenarios are stationary.

**Author Contributions:** Conceptualization, J.A.J.-V. and I.A.; methodology, J.A.J.-V., I.A. and E.T.; software, J.A.J.-V.; validation, J.A.J.-V. and I.A.; formal analysis, J.A.J.-V. and I.A.; investigation, J.A.J.-V., I.A. and E.T.; resources, J.A.J.-V. and I.A.; data curation, J.A.J.-V. and I.A.; writing—original draft preparation, J.A.J.-V. and I.A.; writing—review and editing, J.A.J.-V. and I.A.; visualization, J.A.J.-V., I.A. and E.T.; supervision, I.A.; project administration, I.A.; funding acquisition, J.A.J.-V. All authors have read and agreed to the published version of the manuscript.

**Funding:** This research was funded by Fundación Séneca, Agencia de Ciencia y Tecnología, Región de Murcia, grant number 21271/FPI/19.

**Data Availability Statement:** All data used in the study appear in the manuscript.

**Acknowledgments:** We would like to thank the "Fundación Séneca" for the scholarship awarded to José Antonio Jiménez Valera (21271/FPI/19. Fundación Séneca. Región de Murcia, Spain). Financial support for monitoring field surveys and access to the information in this research were provided by MCT.

**Conflicts of Interest:** The authors declare no conflict of interest.

## Nomenclature

| | |
|---|---|
| $C_c$ | capacitor |
| $c_e$ | volumetric heat capacity of the rock–fluid matrix (J m$^{-3}$ k$^{-1}$) |
| $c_{e,w}$ | volumetric heat capacity of water (J m$^{-3}$ k$^{-1}$) |
| EC | electrical conductivity (mS/cm) |
| g | gravity (m s$^{-2}$) |
| $G_{cd}$ | current generator relative to the heating circuit, placed in the lower half of the cell, parallel to the lower resistance |
| $G_{cu}$ | current generator relative to the heating circuit, placed in the upper half of the cell, parallel to the upper resistance |
| H | vertical depth of the aquifer (m) |
| $J_C$ | current intensity flowing through a capacitor (A) |
| k | permeability (m$^2$) |
| $k_m$ | heat conductivity of the rock–fluid matrix (J/(s m °C)) |
| $K_{Darcy}$ | hydraulic conductivity (m s$^{-1}$) |
| MCT | Mancomunidad de los Canales del Taibilla |
| $R_{cd}$ | resistor placed in the lower half of the cell, relative to heat flow |
| $R_{cu}$ | resistor placed in the upper half of the cell, relative to heat flow |
| T | temperature (°C) |
| $T_z$ | temperature at depth z (°C) |
| $T_{z\ simulated}$ | temperature simulated at depth z (°C) |
| $T_{bottom}$ | temperature at the bottom of the domain (°C) |
| TDPTS | temperature–depth profile time series |
| TI | thermal inertia |
| $T_{ini}$ | initial temperature (°C) |
| $T_{surface}$ | mean surface soil temperature (°C) |
| t | time (s) |
| V | voltage (V) |
| $v_{advection}$ | advective component of velocity (m s$^{-1}$) |
| $v_{bouyancy}$ | velocity caused by buoyancy effects (m s$^{-1}$) |
| $v_z$ | vertical water flow velocity (m s$^{-1}$) |
| $v_{z,upward\ flow}$ | vertical upward water flow velocity component (m s$^{-1}$) |
| $v_{z,recharge\ flwo}$ | vertical downward water flow velocity component (m s$^{-1}$) |
| $v_{z,o,i}$ | inverse problem protocol velocity "i" |
| $v_{z,o,1}$ | inverse problem protocol initial velocity |
| z | vertical spatial coordinate (m) |
| $\alpha$ | thermal diffusivity (m$^2$ s$^{-1}$) |
| $\beta$ | thermal expansion coefficient of water (°C$^{-1}$) |
| $\Delta$ | gradient operator |
| $\Delta T$ | amplitude due to sinusoidal boundary condition at the surface (°C) |
| $\Delta T_{interval}$ | temperature gradient for each depth interval (°C) |
| $(\Delta T)_{z=0}$ | amplitude of the temperature variation at z = 0 (°C) |
| $\Delta v_z$ | groundwater velocity increase (ms$^{-1}$) |
| $\mu$ | water viscosity (kg m$^{-1}$ s$^{-1}$) |
| $\rho_e$ | wet bulk density of the rock–fluid matrix (kg m$^{-3}$) |
| $\rho_{e,w}$ | water density (kg m$^{-3}$) |
| $\tau$ | period of the sinusoidal thermal wave (s) |
| $\varphi$ | mathematical function |

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
