# Peer review of "Quantification of Groundwater Vertical Flow from Temperature Profiles: Application to Agua Amarga Coastal Aquifer (SE Spain) Submitted to Artificial Recharge"

_water, doi:10.3390/w15061093_

Round 1

Reviewer 1 Report

The paper uses temperature profiles to identify the vertical flow component of subsurface flow. 1 D heat transfer (convection-conduction) equation is solved numerically. An inverse approach procedure is applied to fit numerical predictions and observation data. The topic is interesting, but the paper is not well-written. The objectives are not clear and the title does not reflect the content. As presented, the paper does not show any novelty because inverse approach is standard and data used for it are standard. English is not good and should be revised. Also there is a strong assumption by neglecting buoyancy effects. This assumption should be justified. I am hesitating between rejection and major revisions, but I would suggest major revisions because the results are interesting. I invite the author to revise seriously the paper and to re-structure it to show clearly the novelty. As explained in my comments below, the novelty could be related to the site. This can be confirmed by developing a review about existing studies on the site.

- In the abstract: “From 2008, groundwater abstractions along the coastline to supply the Alicante I 10 and II desalination plants were combined with a recharge program carried out to reduce the impact 11 on piezometry and surface ecosystems, consisting in seawater irrigation over the salt marsh.” This sentence does not read well and it is too long.

-“Information related to monthly temperature depth profiles in a piezometer located in the salt marsh, seawater irrigation volumes, rainfall and average air and seawater temperatures was gathered.”  I think it is better to first talk about objectives and then you talk about data in the methodology

- “to the analysis of TDPTS and to the quantification of groundwater vertical flow components“. What is TDPTS?

- From the abstract, I can say that the title of the paper doesn’t reflect the content. The title indicates that the goal of the work is to investigate the effect of recharge on temperature while the abstract explains that data are used to identify vertical groundwater flow.

-“The physics basis of that is the coupled process implied in the flow and heat transfer equation for porous media [7-10].” Could be: The physics basis of that are the coupled processes of flow and heat transfer due to convection phenomenon [7-10].”

-Please cite Fahs et al. 2019 as new reference for the heat transfer in porous media.

Fahs et al. 2019: 10.1007/s11242-019-01356-1

-“Suzuki [7] was the 27 first author who proposed an analytical solution in aquifers, under harmonic temperature 28 boundary conditions and constant vertical flow. Bredehoeft & Papadopulos [ 8] simplified 29 Suzuki’s solution for the case of constant temperature at the soil surface. Lu and Ge [10] 30 proposed analytical solutions for oblique flow considering the severe hypothesis of con31 stant horizontal temperature gradients at all points of the aquifer.”

I do not understand why the authors discuss in this section the analytical solution. The objective of this paper is not related to analytical solution. This analytical solution is used as tool. Thus, it should be discussed later in methodology after providing the context and objective of this study.

I cannot follow the introduction. The objectives of this paper are not clear. After reading the introduction, I can understand that the paper is suggesting a technique for the estimation of vertical subsurface flow component from temperature profiles. Agua Amarga coastal aquifer is selected as example of application. The content of the paper is not at all related to the tile. The introduction should be re-organized. I think it can first discuss application of temperature as indicator of flow. And then mention that usually this is performed with analytical solutions.

I cannot see any novelty in the methodology in the paper. Standard inverse approach is applied. Standard numerical model is developed. Standard data are used. But the methodology could be in the application to the case study. In this case, comparison with other studies on the site should be discussed. If there is no novelty in the methodology, the novelty should be in the application to the site. Thus, the paper should not be presented as a new method applied for the site. Instead, the site should be the first objective and the novelty will be related to it.  

I do not know why the authors are complicating the numerical solution. This is a simple 1D heat transfer equation. It can be solved with 30 lines python code with finite difference.

The results are interesting. However, there is a strong assumption by neglecting buoyancy effect. With a temperature gradient more than 6 °C the effect of buoyancy becomes important. This assumption should be justified.

The paper is not well written. Some terms are not clear: Examples:

“boundary conditions of the unknown parameters in a problem”

Author Response

Dear Reviewer, Thank you very much for your review. The responses to each of the comments/suggestions are in the attached word. 

Sincerely,

The authors. 

Reviewer 2 Report

1. Please add a conceptual map of artificial recharge and explain it in detail. If seawater is used as artificial recharge water, groundwater quality will deteriorate. The reason and purpose of artificial recharge should be clearly presented. It would be good to present a cross section in Figure 1 and explain the artificial recharge concept.

2. If artificial recharge is applied to the surface, the groundwater flow will flow from top to bottom, but the mechanism of going up from bottom to top needs to be explained. Please make sure that this phenomenon does not occur only at point P-8.

3. In terms of methodology, inverse calculation of temperature data is difficult to see as a new concept. Please suggest the difference of the protocol presented in this study from previous studies.

4. In Figure 8 and the boundary condition, check if the groundwater flow can be inverted without considering the temperature at the bottom.

5. It seems that the methodology presented in this paper can be applied limited to the saturation zone. By the way, isn't the field situation a situation where the unsaturated and the saturated zone must be considered at the same time?

Author Response

(The authors gave the same response as above.)

Round 2

Reviewer 2 Report

The questions asked in the article were well explained, and it was judged sufficient to be published in the journal.

Author Response

Thank you very much for your revision.